# IL-32γ Induced Autophagy Through Suppression of MET and mTOR Pathways in Liver Tumor Growth Inhibition

**DOI:** 10.3390/ijms252111678

**Published:** 2024-10-30

**Authors:** Ji-Won Seo, Yong-Sun Lee, In-Sook Jeon, Ji-Eun Yu, Jun-Sang Yoo, Ja-Keun Koo, Dong-Ju Son, Jae-Suk Yoon, Sang-Bae Han, Do-Young Yoon, Yoon-Seok Roh, Jin-Tae Hong, Jung-Hyun Shim

**Affiliations:** 1College of Pharmacy and Medical Research Center, Chungbuk National University, 194-31, Osongsaengmyeong 1-ro, Osong-eup, Cheongju-si 28160, Chungbuk, Republic of Korea; never1858@naver.com (J.-W.S.); isjeon@chungbuk.ac.kr (I.-S.J.); dbwnstkd8482@naver.com (J.-S.Y.); pheej1@naver.com (J.-K.K.); sondj1@chungbuk.ac.kr (D.-J.S.); jyun@chungbuk.ac.kr (J.-S.Y.); shan@chungbuk.ac.kr (S.-B.H.); ysroh@chungbuk.ac.kr (Y.-S.R.); 2Ministry of Food and Drug Safety, 187, Osongsaengmyeong 2-ro, Osong-eup, Heungdeok-gu, Cheongju 28159, Chungbuk, Republic of Korea; kallintz123@korea.kr; 3College of Pharmacy, Mokpo National University, Muan 58554, Jeonnam, Republic of Korea; jieunyu@mnu.ac.kr; 4Department of Biomedicine, Health & Life Convergence Sciences, BK21 Four, Biomedical and Healthcare Research Institute, Mokpo National University, Muan 58554, Jeonnam, Republic of Korea; 5Department of Bioscience and Biotechnology, Konkuk University, Seoul 05029, Republic of Korea; ydy4218@konkuk.ac.kr

**Keywords:** IL-32γ, liver cancer, autophagy, MET, mTOR

## Abstract

Interleukin-32γ (IL-32γ) has diverse functions in various malignancies. In this study, we investigated the role of IL-32γ in autophagy induction in liver cancer cells and delineated the underlying mechanisms. We found that the increased IL-32γ expression inhibited the growth, cell cycle progression, and migration of HepG2 and Hep3B cell lines; it also decreased the expression of related proteins. Furthermore, the IL-32γ overexpression induced autophagy, as indicated by the number of puncta, the expression of LC3, and the expression of autophagy-related markers. The expression levels of LAMP1, a protein essential for autophagosome formation, and colocalization with LC3 also increased. Big data analysis revealed that the expression of MET, a well-known target of autophagy, and the expression of mTOR and mTOR-related proteins were decreased by the IL-32γ overexpression. The combination treatment of MET inhibitor, cabozantinib (2 µM), and IL-32γ overexpression further increased the number of puncta, the colocalization of LC3 and LAMP1, and the expression of autophagy-related proteins. In vivo, liver tumor growth was suppressed in the IL-32γ-overexpressing mouse model, and autophagy induction was confirmed by the increased expression of LC3 and LAMP1 and the decreased expression of autophagy pathway markers (MET and mTOR). Autophagy was also decreased in the liver tumor sample of human patients. ROC curve and spearman analysis revealed that the expression levels of LC3 and IL-32γ were significantly correlated in human tumor serum and tissues. Therefore, IL-32γ overexpression induced autophagy in liver tumors through the suppression of MET and mTOR pathways critical for tumor growth inhibition.

## 1. Introduction

Interleukin-32γ (IL-32γ), originally identified as natural killer cell transcript 4 (NK4), was cloned as an IL-18-induced gene and later renamed IL-32 in 2005 [1]. The IL-32 gene is located on human chromosome 16p13.3, comprising eight exons with the ATG codon in exon 2. Nine small alternatively spliced IL-32 isoforms, namely, IL-32α, IL-32β, IL-32γ, IL-32δ, IL-32ε, IL-32ζ, IL-32η, IL-32θ, and IL-32, are expressed in various tissues, organs, and cell types [2,3,4]. However, their functional differences remain unclear. IL-32 stimulates the production of natural killer (NK) cells and enhances the activity of inflammatory mediators, such as TNF, IL-1β, and IL-6, in addition to white blood cells [5]. Its expression is induced not only in immune tissues but also in various epithelial cells, including those in epithelial cancers [6]. Recent studies have highlighted the involvement of IL-32 in cancer development, and its expression is identified in various cancers such as gastric, lung, and pancreatic cancer [6,7,8,9]. IL-32 in cancer cells exerts anti-cancer effects by inhibiting cell growth and inducing cancer cell apoptosis [10,11,12]. The IL-32γ overexpression inhibits the growth and development of lung tumors by increasing the TIMP-3 expression [13]. It also prevents the growth of lung cancer stem cells by downregulating STAT5 through ITGAV [14]. In colon cancer, the IL-32γ overexpression inhibits cancer cell growth by inducing the inhibition of NF-κB and STAT3 signaling and the reduction of inflammatory cytokine production [5]. However, the implications of IL-32γ for autophagy in liver cancer remain unclear.

Autophagy is a critical cellular physiological process that occurs within cells and regulates the function of cells by breaking down and recycling components [15,16,17]. Through autophagy, macromolecules, aggregated proteins, and damaged organelles are delivered to lysosomes, where they are degraded by lysosomal hydrolases, thereby generating nucleotides, amino acids, fatty acids, sugars, and ATP [18]. In cancer, the loss of autophagy may increase the propensity of cells toward oncogenic transformation because autophagy-deficient cells are often more tumorigenic than normal cancer cells; furthermore, it causes damaged DNA accumulation, genomic instability, and inflammation persistence [19]. Additionally, TGM2 inhibition by PRKCD induces the excessive activation of Beclin1-dependent autophagy; as a result, cell death occurs, and metastasis becomes inhibited in pancreatic cancer [20]. The betulin overexpression induces autophagy through PI3K/AKT/mTOR and AMPK pathways, thereby inducing autophagy and inhibiting metastasis in colon cancer cell lines [21]. Autophagy is critical for apoptosis induction and thus controls cancer growth. It is functionally linked to apoptosis in colon and liver cancer via a common signaling pathway. During autophagy, Bcl-2 implicated in the two processes interacts with Beclin1, which controls autophagy [22,23]. Furthermore, autophagy and apoptosis suppress cancer cell growth by regulating balance through the p53 and JNK MAPK pathways [24].

Cytokines participate in cancer development in inflammatory or immune diseases [25]. They can induce or suppress autophagy [26]. The interplay between autophagy and cytokines may be a fundamental mechanism coordinating the activity of not only innate and adaptive immune systems but also cancer progression [27]. IFN-γ induces autophagy by activating macrophages through pathways related to family M member 1 GTPase Irgm1/IRGM1 [27]. It also promotes autophagy induction in cervical cancer cells via IDO1 expression and kynurenine metabolism [28]. Similarly, TNF-α induces autophagy via the JNK pathway in gastric cancer cell lines. IL-17 increases the activity of the ubiquitin-proteasome system, triggers ERK1/2 phosphorylation to upregulate p62 and Beclin-1 expression, and increases LC3-II accumulation to induce autophagy in liver cancer cell lines [29]. IL-6 induces BECN1 phosphorylation via the JAK2 pathway in colon cancer cells and induces autophagy [30]. The production and secretion of cytokines, including IL-1, IL-18, and TNF-α, can be regulated by autophagy [27,31]. TGF-β induces the TFEB expression via the canonical Smad pathway in Smad4-positive cells and facilitates TFEB-mediated autophagic activation in pancreatic cancer cell lines [32]. IL-12 induces autophagy in breast cancer cell lines via AMPK and PI3K/Akt pathways [33]. Moreover, SOCS5 inhibition increases the expression of autophagy-related proteins, thus preventing cancer metastasis [34].

Studies have reported that the PI3K/AKT/mTOR pathway is involved in autophagy regulation. The overexpression of TGF-β, an immunosuppressive cytokine, stimulates autophagy by inhibiting the mTOR pathway [35]. IL-37 induces the expression of autophagy through the decreased AMPK-mTOR pathway in inflammatory diseases [36]. The overexpression of CCL2 that promotes the expression of the PI3K/AKT/mTOR pathway decreases the expression of autophagy in non-small cell lung cancer [37]. However, the role of IL-32γ in autophagy in cancer cell growth has not been reported yet.

In this study, we investigated whether the IL-32γ overexpression regulated autophagy in liver cancer cells and its basic molecular mechanisms.

## 2. Results

### 2.1. IL-32γ Overexpression Inhibited Liver Cancer Cell Growth

In previous studies, the IL-32γ overexpression inhibited tumor growth in lung and colon cancer. We observed that the IL-32γ expression is higher in the liver compared with other organs. Thus, we investigated whether IL-32γ expression suppresses liver cancer cell growth. We investigated the effect of IL-32γ overexpression on the growth of liver cancer cells and found that IL-32γ overexpression inhibited their growth, with an IC_50_ value of 1.76 µg in HepG2 cells and 1.55 µg in Hep3B cells for 24 h (Figure 1A). Next, we examined the markers related to cell growth and migration via western blotting and observed that the IL-32γ overexpression decreased the expression of related markers (MMP13, MMP9, CDK5, CDK3, Cyclin E1, Cyclin D1, and PCNA) in HepG2 and Hep3B cell lines in a concentration-dependent manner (Figure 1B).

### 2.2. IL-32γ Overexpression Induced Liver Cancer Cell Apoptosis

One of the mechanisms for inhibiting cancer cell growth is the induction of apoptosis. In previous studies, it was reported that an IC_50_ value (approximately 1.7 µg/mL) induced apoptosis in colon cancer cells (SW620, 52% and HCT116, 45%). In lung cancer cells, the IC_50_ value (approximately 0.8 µg/mL) induced apoptosis (LLC, 61% and A549, 55%). We evaluated the extent of apoptosis in liver cancer cells to see whether the induction of apotosis a similar pattern to cell growth inhibition may be. A TUNEL assay of the effect of the IL-32γ overexpression on the induction of apoptosis revealed that its overexpression significantly increased the apoptosis of HepG2 and Hep3B cells. The number of apoptotic HepG2 and Hep3B cells was 25% and 33%, respectively, indicating that apoptosis in the liver cancer cells may be less significant in inhibiting the liver cancer cell growth than lung and colon cancer cells. Western blot analysis of the expression of apoptosis-related proteins showed that the IL-32γ overexpression increased the expression of apoptosis-related proteins (Bax and Caspase 3) in the liver cancer cell lines. Conversely, it decreased the Bcl-2 expression (Figure 1D). Therefore, the IL-32γ overexpression induced apoptosis in liver cancer cell lines.

### 2.3. IL-32γ Overexpression Induced Autophagy in Liver Cancer Cell Lines

The association between IL-32γ and autophagy has not been clearly established. We also wondered whether autophagy could be significant in the IL-32γ overexpression-dependent inhibition of cancer cell growth. Therefore, we investigated the relationship between IL-32γ overexpression and autophagy in liver cancer cells. Specifically, we determined whether the IL-32γ overexpression induced LC3 puncta formation through fluorescent staining of HepG2 and Hep3B liver cancer cells with the LC3 antibody. We found that the IL-32γ overexpression significantly increased the number of LC3 puncta that accumulated in liver cancer cells (Figure 2A). Autophagy is a degradation process that occurs after autolysosomes and autophagosomes fuse with lysosomes. Therefore, we investigated colocalization with the late endosome/lysosome marker protein LAMP1 to further investigate autophagy induction by IL-32γ overexpression. Our findings revealed that the IL-32γ overexpression enhanced the colocalization of LC3 and LAMP1, consequently inducing autophagy. Our results indicated that IL-32γ overexpression increased the LAMP1 expression and promoted lysosomal recruitment with autophagosomes, thereby increasing autophagy fusion (Figure 2B). Western blotting showed that the IL-32γ overexpression increased the expression of autophagy-related markers (LC3, Atg5, Beclin1, and p62) (Figure 2C). Thus, IL-32γ overexpression induced autophagy in the liver cancer cell lines.

### 2.4. IL-32γ Overexpression Induced Autophagy Through the MET Pathway

In cancer, autophagy can be induced through various autophagy pathways. However, the specific relationship between IL-32γ and autophagy pathways has yet to be clearly defined in liver cancer. We determined the proteins that could be associated with IL-32γ-induced autophagy through the STING big data analysis tool. We found that the top five proteins (MET, PRKCD, IL18RAP, CXCL8, and TNF) were associated with IL-32γ (Figure 3A). We then investigated the expression of the target and observed that MET was significantly decreased by the IL-32γ overexpression. RT-qPCR confirmed that the MET expression decreased the most when IL-32γ was overexpressed (Figure 3B). Therefore, MET was highly associated with IL-32γ. To further examine the involvement of the MET pathway with IL-32γ-induced autophagy in liver cancer cell lines, we examined the combination effects of Cabozantinib, a MET inhibiting compound, and IL-32γ overexpression on autophagy induction in liver cancer cell lines. We verified the expression of LC3 puncta in liver cancer cell lines through immunocytochemistry. Our results revealed that the combination treatment of Cabozantinib and IL-32γ overexpression increased much more than the number of puncta as well as expression of LC3 in liver cancer cell lines (Figure 3C). Immunocytochemistry confirmed the colocalization of LC3 and LAMP1 after the combination treatment of Cabozantinib and IL-32γ overexpression. This combination treatment also further induced a high expression of LC3 and LAMP1 colocalization in liver cancer cell lines (Figure 3D). Western blotting demonstrated that such treatment further increased the expression of autophagy-related proteins (LC3, Atg5, Beclin1, and p62) and decreased the MET expression (Figure 3E). In this study, this combination treatment induced further autophagy in liver cancer cell lines.

### 2.5. MET Pathway Associated with mTOR Pathway

IL-32γ overexpression induced autophagy via the MET pathway. However, the pathways or mechanisms in the MET pathway-dependent autophagy are not yet clearly understood. We thus further examined the relationship between MET and key autophagy pathways. We used the STRING big data analysis tool to determine the association between MET and autophagy-related proteins (mTOR, TP53, PIK3CA, and AKT1). We found the top four autophagy-related proteins highly associated with MET (Figure 4A). RT-qPCR showed that the expression level of mTOR among the four proteins was the most significantly decreased by the IL-32γ overexpression (Figure 4B). STRING big data analysis revealed mTOR-related proteins (PDPK1, EIF4EBP1, and ULK1; Figure 4C). Western blotting indicated that the expression levels of mTOR-related proteins (p-mTOR, p-4E-BP1, p-S6 kinase, and p-ULK1) and MET proteins were decreased by the IL-32γ overexpression, but p-ULK1 was increased (Figure 4D). These data suggested that the IL-32γ overexpression induced autophagy-associated with MET and mTOR pathways. Western blotting verified that the combination treatment of Cabozantinib and IL-32γ overexpression significantly decreased the mTOR-related proteins (p-mTOR, p-4E-BP1, and p-S6 kinase) and MET but increased p-ULK1 in the liver cancer cell lines (Figure 4E). Therefore, autophagy was increased by the combination treatment of Cabozantinib and IL-32γ overexpression through MET and mTOR pathways.

### 2.6. IL-32γ Overexpression Inhibited Liver Cancer Cell Growth and Induced Autophagy In Vivo

In previous studies, the IL-32γ overexpression mouse significantly reduced the growth of lung and colon cancers in a Xenograft mouse mode. On the basis of these studies, we determined whether IL-32γ overexpression also inhibits liver cancer growth in a similar pattern. The day after the subcutaneous injection of HepG2 cells, we monitored and calculated tumor size twice for 3 weeks. The volume and weight of tumor were dramatically decreased in the IL-32γ-overexpressing mouse model (Figure 5A). We then investigated whether the induction of autophagy could be related to tumor growth inhibition in vivo. We investigated the induction of autophagy and pathways. Immunohistochemistry showed that MET and mTOR expression levels decreased but autophagy-related markers (LC3, and LAMP1) increased in the IL-32γ-overexpressing mouse liver tumor tissues (Figure 5B). We also examined the colocalization with LC3 and LAMP1 in the control and IL-32γ-overexpressing mouse tumor tissues through immunofluorescence analysis. Our results indicated that the colocalization with LC3 and LAMP1 was more increased in the control mouse tumor tissues than in the IL-32γ-overexpressing mouse tissues (Figure 5C). We also evaluated the colocalization with LAMP1 and MET through immunofluorescence and found that the LAMP1 expression was more increased than that in the control; furthermore, the MET expression was more decreased than that in the control. We checked that autophagy was induced by the colocalization with LAMP1 and MET in an IL-32γ-overexpressing mouse (Figure 5D). Through western blotting, we found that the expression levels of cell growth and migration-related markers (MMP3, and PCNA) and autophagy-related markers (MET and p-mTOR) were decreased; conversely, the expression levels of cleaved caspase 3, LC3, and LAMP1 were increased in the IL-32γ-overexpressing mouse liver tumor tissues (Figure 5E). Therefore, the IL-32γ overexpression induced autophagy and suppressed liver tumor growth in the in vivo xenograft model.

### 2.7. Autophagy, MET, and mTOR Pathways Were Associated with Human Liver Cancer

Based on the experimental results obtained from both cell and animal models. We investigated the association between IL-32γ and autophagy in human liver cancer patient samples. We analyzed the change in the expression of cell growth and apoptosis-related markers (MMP3, PCNA, and Caspase 3) and autophagy-related proteins (LC3, LAMP1, p-mTOR, and MET) in human liver tumor tissues. The expression levels of apoptosis-related proteins (Caspase 3) and autophagy-related proteins (LC3 and LAMP1) were more decreased in liver tumor tissues than in non-tumor tissues, but the expression levels of cell growth markers (MMP3 and PCNA), MET, and p-mTOR were increased in liver tumor tissues compared with those in non-tumor tissues (Figure 6A). We also examined the expression of autophagy-related markers in liver tumor tissues through immunohistochemistry. Furthermore, the expression levels of autophagy-related markers (LC3 and LAMP1) were more increased in non-tumor tissues than in liver tumor tissues, but the expression levels of mTOR and MET were decreased in non-tumor tissues (Figure 6B).

### 2.8. Relationship Between IL-32γ and Autophagy

Next, receiver operating characteristic (ROC) curve analysis was conducted in the serum of patients with liver tumors and non-tumors to see whether IL-32γ overexpression could be associated with induction of autophagy. The levels of LC3 in the serum (389.33 ± 187.327 pg/mL) and tissue (236.4 ± 35.815 pg/mL) of the tumor group and the levels of IL-32γ were 30.1 ± 8.6 pg/mg in the serum and 46.1 ± 8.5 pg/mg in the tissue in the tumor group (Figure 6C,D). The ROC curve analysis of LC3 showed 90% sensitivity, 90% specificity, 615.7 pg·mL^−1^ cutoff value, and 0.95 area under the curve (AUC). We also found that IL-32γ showed 50% sensitivity, 100% specificity, 27.9 pg·mg^−1^ cutoff value, and 0.81 AUC (Figure 6C). Additionally, the ROC curve analysis of LC3 and IL-32γ in tissues showed 70% sensitivity, 90% specificity, 256.3 pg·mL^−1^ cutoff value, and 0.89 AUC for LC3 and 100% sensitivity, 90% specificity, 53.78 pg·mg^−1^ cutoff value, and 0.93 AUC for IL-32γ (Figure 6D). We performed the Spearman correlation test to evaluate the correlation score between IL-32γ and LC3 in liver tumor patients. The correlated score with IL-32γ and LC3 in tumor serum showed the 0.8126 of R value and *p* < 0.0043; the score of the tissue indicated the 0.7118 of R value and *p* < 0.021 (Figure 6E). Thus, IL-32γ and LC3 were significantly correlated in liver tumor serum and tissues during liver cancer development.

## 3. Discussion

Several studies have reported that the IL-32γ overexpression suppresses various cancer cell lines [5,14,38]. Overexpression of IL-32γ overexpression induced the apoptosis of colon cancer cells [39]. Furthermore, induced autophagy inhibits cancer cell growth and increases apoptosis [40,41,42,43]. On the basis of these results, we investigated whether IL-32γ overexpression induces autophagy in liver cancer and possible understanding mechanisms.

In this study, we investigated the potential target proteins interacting with IL-32γ in the autophagy of liver cancer. The analysis of the STRING big database revealed that IL-32γ was highly associated with MET. MET is a proto-oncogene located on chromosome 7q31.2 that encodes a receptor tyrosine kinase [44]. MET and its ligand, HGF, participate in signaling pathways involved in oncogenic processes, including cell proliferation regulation, invasion, angiogenesis, and cancer stem cell regulation [45]. Increased MET and HGF levels have been reported in various human cancers [46]. In other papers, reduced MET protein levels inhibit cancer cell growth and induce apoptosis in various cancer cells [47,48,49,50]. MET decreased by dictamnine suppresses lung cancer cell growth by downregulating PI3K/AKT/mTOR and MAPK signaling pathways [51]. Cabozantinib, which is a MET inhibitor, inhibits tyrosine kinases, including vascular endothelial growth factor receptors 1, 2, and 3, MET, and AXL, which are implicated in liver cancer progression [52]. We also found that the IL-32γ overexpression enhanced autophagy induction. Cabozantinib enhances PD-1 activity to induce an immune response [53]. Cabozantinib increases apoptosis in breast, lung, and glioma tumor models and decreases tumor and endothelial cell proliferation with the suppression of liver cancer cell growth [54]. The combination of Cabozantinib and IL-32γ overexpression further increased autophagy. In vivo, Cabozantinib induces autophagy in xenograft and PDX mouse models [55]. In the present study, we found the high association between IL-32γ and MET through the STRING big database. Additionally, the IL-32γ overexpression decreased the MET expression in liver cancer cell lines. IL-32γ overexpression also reduced the MET expression in liver cancer cell lines by the combination of Cabozantinib with IL-32γ overexpression and further induced autophagy. Thus, IL-32γ induced autophagy by inhibiting the MET pathway.

mTOR, a serine/threonine kinase, regulates cell growth and proliferation in response to various signals and participates in autophagy regulation; it has been implicated in diabetes mellitus, neurodegenerative diseases, and various cancers [56,57]. In another study, rapamycin, an mTOR inhibitor, affects the downstream inactivation of S6K1 and 4E-BP1 by inhibiting phosphorylation, which decreases protein synthesis and cell cycle arrest in the G1 phase and suppresses the growth of various cancer cell lines [58]. GSK458, a PI3K/mTOR inhibitor, decreases the mTOR expression and suppresses the tumorigenesis, metastasis, and growth of ovarian cancer cell lines [59]. NLRC5 overexpression improves cardiac hypertrophy by inactivating the mTOR pathway, thereby increasing autophagy [60]. The concentration-dependent treatment of nitidine chloride causes autophagy by inhibiting the Akt/mTOR pathway in ovarian cancer [61]. Also, apigenin, which elicits oxidative and anti-tumor effects, induces autophagy, and inhibits cell proliferation by inhibiting the PI3K/Akt/mTOR pathway [62]. In the present study, we found that MET was associated with mTOR analyzed by the STRING method with big database, and the IL-32γ overexpression reduced the mTOR and mTOR-related proteins (p-mTOR, p-4E-BP1, p-S6 kinase) expression but increased p-ULK1 in liver cancer cell lines. Thus, it is possible that MET pathway could be associated with mTOR pathway. The combination treatment exhibits synergies targeting multiple pathways in breast cancer and other cancers [63,64]. It is also more effective than a single treatment [65,66]. In renal cancer, the combination treatment of Cabozantinib and honokiol induces apoptosis and autophagy to a greater extent than the single treatment does [67]. The combination of Cabozantinib and temozolomide induces autophagy by decreasing the expression of mTOR protein in uterine sarcoma [68]. In colon cancer, the combination treatment of SBI-0206965 and Cabozantinib increased the expression of autophagy-related proteins to a higher degree than the single treatment does [55]. In the present study, we found that the combination treatment of Cabozantinib and IL-32γ overexpression further increased the number of LC3 puncta and colocalization with LC3 and LAMP1 in liver cancer cell lines. In addition, the combination treatment of Cabozantinib and IL-32γ overexpression further increased the expression of autophagy-related markers and p-ULK1; however, the expression levels of MET, p-mTOR, and mTOR-related proteins further decreased in liver cancer cell lines. Therefore, the IL-32γ overexpression induced autophagy by suppressing MET and mTOR pathways in liver cancer cell lines.

In a previous study, we generated IL-32γ transgenic mice [5]. We injected HepG2 cells and monitored the tumor size twice a week for 3 weeks. These results showed that the IL-32γ overexpression mouse model significantly reduced the tumor size and weight compared with those in the control group. In the present study, we found that tumor volume and weight decreased in IL-32γ-overexpressing mouse tumors. Immunohistochemistry revealed that the LC3 and LAMP1 expression increased, but the mTOR and MET expression decreased in IL-32γ-overexpressing mouse tumor tissues. The colocalization with LC3 and LAMP1 increased to a higher extent in the control mouse tumor tissues than in the IL-32γ-overexpressing mouse tumor tissues. Furthermore, western blotting showed that the expression levels of cell growth-related markers (MMP3 and PCNA), p-mTOR, and MET decreased, but the expression levels of apoptosis-related markers (cleaved caspase 3) and autophagy-related markers (LC3 and LAMP1) increased in overexpressing mouse tumor tissues. Therefore, the IL-32γ overexpression not only suppressed cancer cell growth but also increased autophagy in vivo. In patients with liver tumors, western blotting revealed that the expression of apoptosis-related protein (Caspase 3) and autophagy-related proteins (LC3 and LAMP1) decreased in liver tumor tissues; conversely, the expression of cell growth-related markers (MMP3 and PCNA), MET, and p-mTOR increased. Immunohistochemistry showed that LC3 and LAMP1 decreased, but mTOR and MET increased in liver tumor tissues. IL-32γ and LC3 concentrations in the non-tumor group were higher than those in the tumor group of the serum and tissues of the patients with liver tumor. The AUC values obtained by the ROC analysis of IL-32γ were 0.81 in serum and 0.93 in tissues. For LC3, the AUC values were 0.95 in serum and 0.89 in tissues. Moreover, our spearman correlation test showed that the correlated score with IL-32γ and LC3 in tumor serum showed the 0.8126 of R value and *p* < 0.0043; the score of the tissue indicated the 0.7118 of R value and *p* < 0.021. This data also demonstrated a significant correlation between IL-32γ and autophagy in human patient samples.

In conclusion, our study confirmed that IL-32γ overexpression suppressed liver tumor growth by induction of autophagy through the suppression of the MET and mTOR pathways. These findings suggest that IL-32γ may represent a promising therapeutic target for liver cancer treatment.

## 4. Materials and Methods

### 4.1. Cell Culture

HepG2 and Hep3B liver cancer cells were obtained from the American Type Culture Collection (Manassas, VA, USA). The cells were cultured in RPMI 1640 and DMEM/high glucose, supplemented with 10% heat-inactivated fetal bovine serum (FBS, #19421001, Coring, New York, NY, USA), 100 U/mL penicillin, and 100 μg/mL streptomycin. Cell cultures were maintained in an incubator with a humidified atmosphere of 5% CO_2_ at 37 °C.

### 4.2. Transfection

For transfection, HepG2 and Hep3B cells were transiently transfected with Myc-IL-32γ using Lipofectamine 3000 (for plasmid DNA) reagent (#L3000001, Thermo Fisher Scientific, San Jose, CA, USA), according to the manufacturer’s protocol. Transfection was performed using Opti-MEM medium. The cells were incubated for 1–2 days in a humidified atmosphere with 5% CO_2_ at 37 °C.

### 4.3. Cell Lysate Preparation and Western Blotting Analysis

HepG2 and Hep3B cells were washed with cold PBS and harvested with lysis buffer (50 mM Tris-HCl (pH7.6), 0.1% Triton X-100, 0.25 mM NaCl, and 2 mM EDTA with phosphatase and protease inhibitors) and lysed by incubation on ice for 20 min. And cell lysates were centrifuged at 12,000× *g* for 30 min at 4 °C. Equal amounts of protein were separated by 8–15% SDS-PAGE electrophoresis and then transferred to a polyvinylidene fluoride membrane. The membranes were immunoblotted with the following specific primary antibodies; MMP9 (dilution 1:1000; #ab38898, Abcam), MMP13 (dilution 1:1000; #ab39012), Cyclin D1 (dilution 1:1000; #ab134175), CDK3 (dilution 1:1000; #ab96847), LC3B (dilution 1:1000; #ab48394), IL32 (dilution 1:1000; #ab37158), MET (dilution 1:1000; #ab216574, purchased from Abcam, Cambridge, UK), CDK5 (dilution 1:1000; #sc-249), Cyclin E (dilution 1:1000; #sc-377100), PCNA (dilution 1:1000; #sc-9857), Bcl-2 (dilution 1:1000; #sc-7382), Bax (dilution 1:1000; #sc-7480), MMP3 (dilution 1:1000, #sc-21732), PCNA (dilution 1:1000; #sc-9857), β-actin (dilution 1:1000; #sc-47778 purchased from Santa Cruz Biotechnology, Dallas, TX, USA), Beclin1 (dilution 1:1000; #3495S), Atg5 (dilution 1:1000; 2630S), p62 (dilution 1:1000; #5114), p-mTOR (dilution 1:1000; #2971S), p-4E-BP1 (dilution 1:1000; #2855S), p-S6 kinase (dilution 1:1000; #9205S), and p-ULK1 (dilution 1:1000; #5869S), Cleaved caspase 3 (dilution 1:1000; #9661S), and Caspase 3 (dilution 1:1000; #9662S purchased from Cell Signaling Technology, Danvers, MA, USA). β-actin was used for loading control. The blots were incubated with HRP-conjugated secondary antibodies, and the reactions were visualized by using ECL substrate (#WBKLS0500, Millipore, Billerica, MA, USA) and detected with a FUSION Solo S chemiluminescence detection system (Vilber Lourmat, Collégien, France).

### 4.4. Immunocytochemistry

Immunocytochemistry was performed as previously described [69]. HepG2 and Hep3B cells were cultured on coverslips and transfected. The cells were washed with PBS and fixed using 4% paraformaldehyde for 15 min at room temperature. Cells were then permeabilized with cold methanol for 5 min and blocked with 4% BSA in PBS containing 0.1% Triton X-100 (PBS-T) for 1 h. The primary antibodies with appropriate dilutions were added, and the cells were incubated overnight at 4 °C. After washing three times with PBS-T, coverslips were incubated with Alexa Fluor 488 (#A32723, #A32731, Invitrogen, Carlsbad, CA, USA) or Texas Red (#T-862, #T-2767, Invitrogen)-conjugated secondary antibodies for 1 h at room temperature. The cells were then incubated with 1 μg/mL DAPI (#D9542, Sigma-Aldrich, St. Louis, MO, USA) for 5 min at room temperature and subsequently mounted using the Fluoromount-G Mounting Medium (#0100-01, Southern Biotech, Birmingham, AL, USA). Cells were visualized using the ZEISS Axio Observer fluorescence microscope system (Carl Zeiss, Oberkochen, Germany). Digital images were analyzed using ZEN 2.1 software (Carl Zeiss, Version 2.1). For quantitative analysis, 50 cells were counted in 10 random fields and performed three independent experiments.

### 4.5. Immunohistochemistry

Immunohistochemistry was performed as described previously [18]. The Control and IL-32γ overexpression mice tumor tissue sections were blocked with 3% normal goat serum for 30 min; the sections were then incubated with antibodies for MMP3 (dilution 1:100, #sc-21732), PCNA (dilution 1:100; #sc-9857, purchased from Santa Cruz Biotechnology, Dallas, TX, USA), Cleaved caspase 3 (dilution 1:100; #9661S), p-mTOR (dilution 1:100; #2971S purchased from Cell Signaling Technology, Danvers, MA, USA), LC3B (dilution 1:100; #ab48394), LAMP1 (dilution 1:100; #ab25630), and MET (dilution 1:100; #ab216574, purchased from Abcam, Cambridge, UK) at the appropriate dilution in blocking serum for overnight at 4 °C. The slides were washed in PBS, followed by the avidin–biotin–peroxidase complex (#PK-6101, Vector Laboratories, Burlingame, CA, USA). The slides were washed, and the peroxidase reaction was developed with diaminobenzidine and peroxide (#SK-4100, Vector Laboratories), mounted in Aqua-Mount, and evaluated under a light microscope (Olympus, Tokyo, Japan).

### 4.6. Immunofluorescence

Tumor tissue sections were blocked with 3% normal goat serum for 30 min; the sections were then incubated with antibodies for LC3 and LAMP1 at the appropriate dilution in blocking serum overnight at 4 °C. The slides were washed in PBS and incubated with Alexa Fluor 488 (#A32731, Invitrogen) or Texas Red (#T-862, Invitrogen Carlsbad, CA, USA)-conjugated secondary antibodies for 1 h at room temperature. The sections were treated with a TrueVIEW Autofluorescence Quenching Kit (#SP-8400-15, Vector Laboratories) to minimize non-specific signals. The slides were then incubated with 1 μg/mL DAPI (#D9542, Sigma-Aldrich, St. Louis, MO, USA) for 5 min at room temperature and subsequently mounted (#0100-01, Southern Biotech, Birmingham, AL, USA). The slides were visualized using ZEISS Axio Observer fluorescence microscope system (Carl Zeiss, Oberkochen, Germany). Digital images were analyzed using ZEN 2.1 software (Carl Zeiss, Version 2.1).

### 4.7. TUNEL Assay

According to the manufacturer’s instructions, the DeadEnd^TM^ Fluorometric TUNEL System (Promega, Wisconsin, WI, USA) was used to detect apoptotic cells. HepG2 (1 × 10^4^ cells/well) and Hep3B (1 × 10^4^ cells/well) cells were cultured on 8-chamber slides after being transfected with Myc-IL-32γ. Apoptotic cells were visualized using the ZEISS Axio Observer fluorescence microscope system (Carl Zeiss, Oberkochen, Germany). Digital images were analyzed using ZEN 2.1 software (Carl Zeiss).

### 4.8. RT-qPCR Assay

Total RNA was isolated from tissues and cell samples using the RiboEx Total RNA (#301-001, GeneAll Biotechnology Co., Seoul, Korea) and reverse-transcribed into cDNA using High-Capacity cDNA Reverse Transcription kit (#4368813, Applied Biosystems, Foster City, CA, USA) according to manufacturer’s protocol. Real-time PCR was performed using the SYBR Green PCR Master Mix (#4344463, Applied Biosystems) and analyzed using an ABI PRISM 7700 Sequence Detection system (Applied Biosystems). All primers utilized for RT-qPCR were designed using the ‘Primer3’ on website and purchased from Bioneer Corp. (Daejeon, Republic of Korea). The primers used in the RT-qPCR are presented in Table 1.

### 4.9. Xenograft

Generation of IL-32γ transgenic mice was previously described [5]. A 705 base-pair fragment of the hIL-32γ gene was subcloned into the *EcoRI* sites of the pCAGGS expression vector. IL-32γ insertion was confirmed by amplification of genomic DNA isolated from the transgenic mice tails. Animals were maintained under controlled conditions of temperature and light. They were provided standard mice feed and water ad libitum. For tumor growth model, HepG2 cells [1 × 10^7^ cells/200 μL in phosphate-buffered saline (PBS) with a 21-gauge needle] were injected subcutaneously in control mice and IL-32γ transgenic mice.

The tumor volumes were measured using Vernier callipers, and the following formula was used to determine tumor volumes: (A × B^2^)/2, where A represents the bigger dimension and B represents the smaller dimension. The animals were sacrificed with carbon dioxide inhalants. The tumors were removed from the surrounding muscles and dermis. All protocols involving mice in this study were reviewed and approved by the Chung Buk National University Institutional Animal Care and Use Committee (IACUC) and complied with the Korean National Institute of Health Guide for the Care and Use of Laboratory Animals (CBNUA-2120-23-02).

### 4.10. Human Samples

Human tissues and serum samples from liver cancer patients and normal controls (20 samples, respectively) were obtained from Biobank of Ajou University Hospital, members of Korea Biobank Network. All studies using human samples were conducted in accordance with the Declaration of Helsinki and were approved by the Ethics Committee of Chungbuk National University Medical Center (IRB No. CBNU-202006-0096). The experiments were undertaken with the understanding and written consent of each subject.

### 4.11. ELISA

Concentration of human LC3 and IL-32γ in human serum was measured by ELISA kit (#MBS1603826 and #MBS751394, MyBioSource, SanDiego, CA, USA) according to manufacturer’s protocol.

### 4.12. Big Data Analysis

Protein-protein interaction of IL-32γ, MET, and mTOR-related proteins was analyzed using STRING (https://string-db.org/), accessed on 1 January 2024 [70].

### 4.13. Statistical Analysis

Statistical analyses were performed using GraphPad Prism 3 (GraphPad Prism 8.4.2) (La Jolla, CA, USA). Error bars represent standard deviations (SDs), unless indicated otherwise. Pairwise comparisons were performed using the student’s *t*-test. Multiple comparisons were performed using one-way analysis of variance followed by Tukey’s test. Differences with *p* values lower than 0.05 were considered statistically significant when *p* values were lower than 0.05.

## Figures and Tables

**Figure 1 ijms-25-11678-f001:**
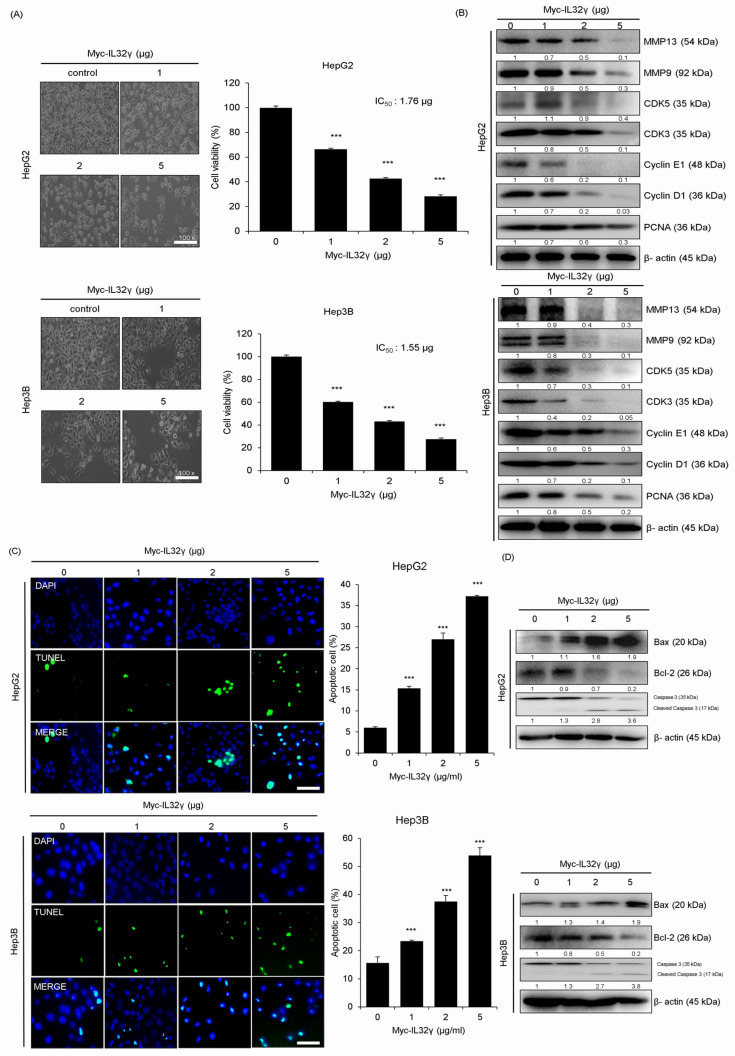
Effect of IL-32γ overexpression on the liver cancer cell growth and apoptosis. HepG2 and Hep3B cells were transfected with IL-32γ (0, 1, 2, and 5 μg) for 24 h. (**A**) Representative cell image of HepG2 and Hep3B cells. Cell viability of IL-32γ overexpression in HepG2 and Hep3B cells was calculated. (**B**) The expression of migration and cell cycle-associated proteins was determined by western blot analysis. Data are presented as mean ± S.D. from three independent experiments. ***, *p* < 0.001 indicates statistically significant differences from the control group. (**C**) The apoptotic index was determined as the TUNEL-positive cell number/total DAPI-stained cell number. Scale bar, 100 μm. Data are presented as mean ± S.D. from three independent experiments ***, *p* < 0.001 indicates statistically significant differences from the control cells. (**D**) The expression of apoptosis-associated proteins determined by Western blot analysis.

**Figure 2 ijms-25-11678-f002:**
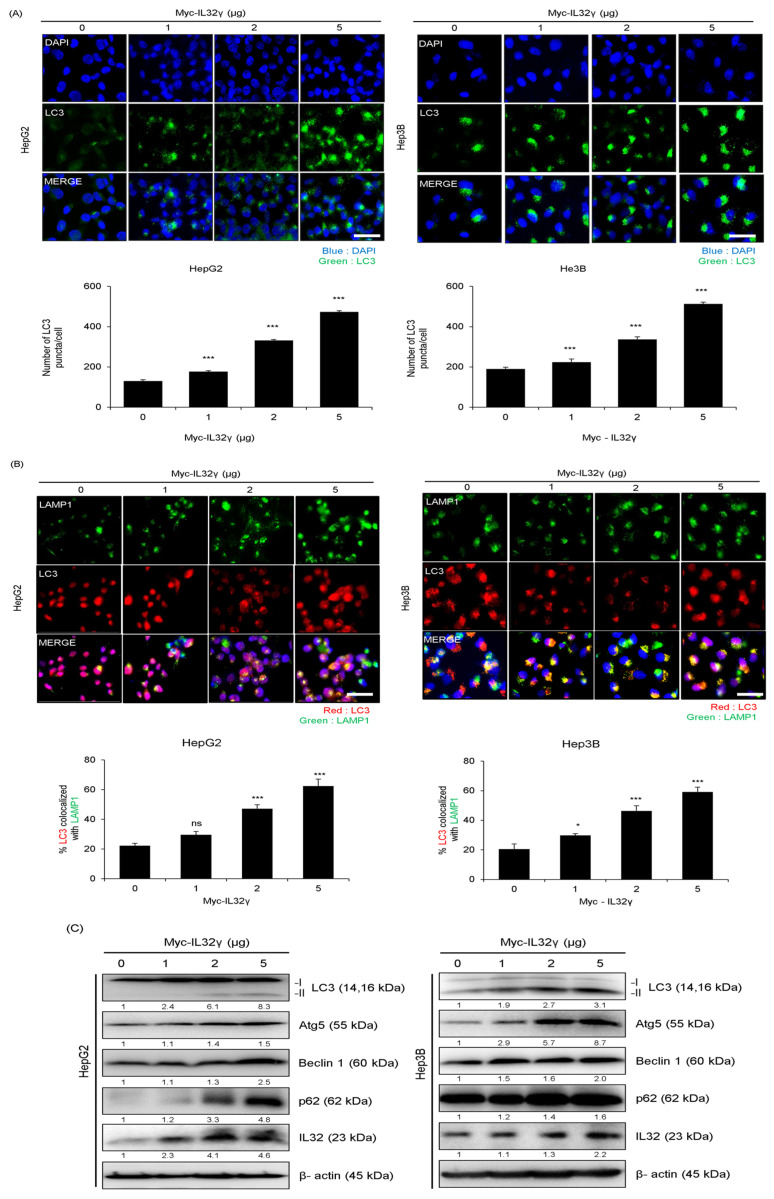
Effect of IL-32γ overexpression on liver cancer autophagy. (**A**–**C**) HepG2 and Hep3B cells were transfected with IL-32γ (0, 1, 2, and 5 μg) for 24 h. (**A**) Transfected cells were stained with LC3 antibody. LC3 puncta formation was using fluorescent microscopy. The number of LC3 puncta per cell was calculated. The data was the average of three independent experiments, and error bars were mean ± SD. ***, *p* < 0.001. Scale bar, 100 μm. (**B**) The cells were fixed and permeabilized. Cells were immunostained with LAMP1 (green) and LC3 (red) during fusion with autophagosomes and lysosomes. The cell nucleus was stained with DAPI (blue). Autolysosome localization was observed by fluorescence microscopy. The data was the average of three independent experiments, and error bars indicate the mean ± SD. *, *p* < 0.05, ***, *p* < 0.001. ns, not significant Scale bar, 100 μm. (**C**) The expression of autophagy-associated proteins determined by western blot analysis.

**Figure 3 ijms-25-11678-f003:**
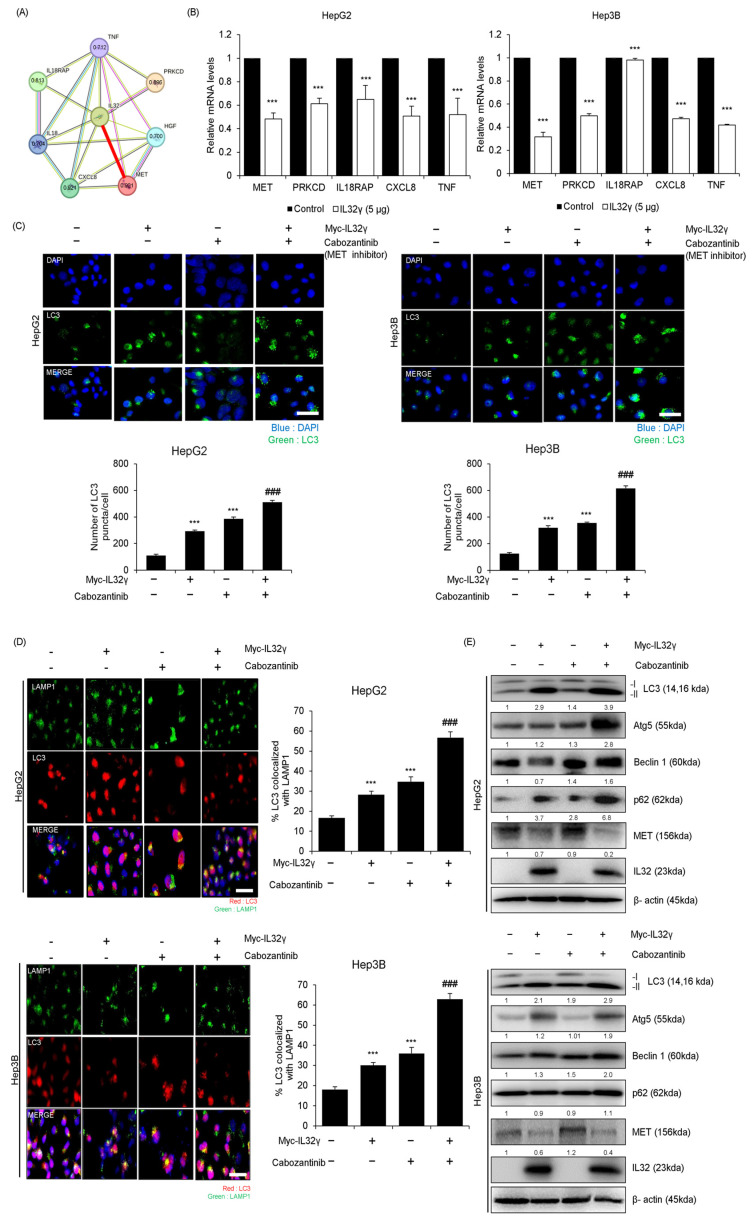
Association between IL-32γ and MET pathway. (**A**) Protein-association network analysis of IL-32γ and MET and its predicted target proteins using the STRING database. The red line represents the robust interaction between IL-32 and MET. This value means a combined score with IL-32 and MET, its predicted target proteins in the STRING database. (**B**) HepG2 and Hep3B cells were transfected with IL-32γ (5 μg) for 24 h. The mRNA level of *MET*, *PRKCD*, *IL18RAP*, *CXCL8*, and *TNF* was measured by RT-qPCR. Data are presented as mean ± S.D. from three independent experiments. ***, *p* < 0.001 indicates statistically significant differences from the control group. (**C**) HepG2 and Hep3B cells were transfected with IL-32γ (2 μg) or Cabozantinib (2 μM) for 24 h. Transfected cells were stained with LC3 antibody. Using fluorescent microscopy, LC3 puncta formation was detected. The number of LC3 puncta per cell was calculated. The data was the average of three independent experiments, and error bars were mean ± SD. ***, *p* < 0.001. Scale bar, 100 μm. ### indicates statistically significant differences from the Cabozantinib or IL-32γ group. (**D**) The cells were fixed and permeabilized. Cells were immunostained with LAMP1 (green) and LC3 (red) during fusion with autophagosomes and lysosomes. The cell nucleus was stained with DAPI (blue). Autolysosome localization was observed by fluorescence microscopy. The data was the average of three independent experiments, and error bars were mean ± SD. ***, *p* < 0.001. Scale bar, 100 μm. ### indicates statistically significant differences from the Cabozantinib or IL-32γ group. (**E**) The expression of autophagy-related proteins determined using western blotting.

**Figure 4 ijms-25-11678-f004:**
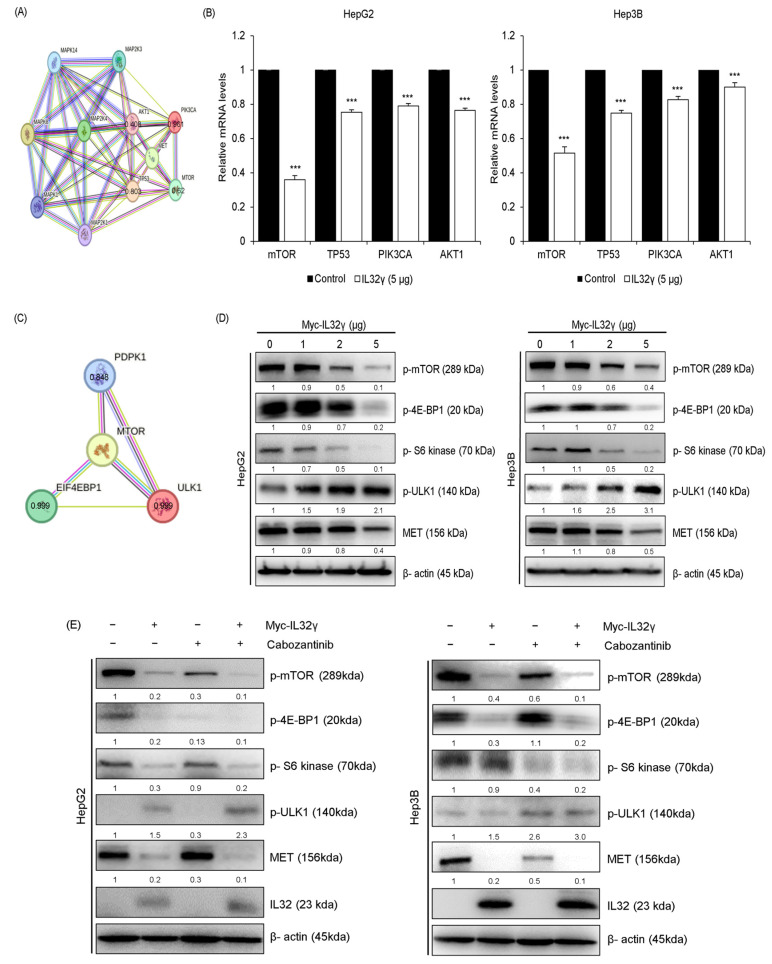
Association between MET pathway and mTOR pathway. (**A**,**C**) Protein-association network analysis of MET and mTOR, mTOR-related proteins, and its predicted target proteins using the STRING database. The red line represents the robust interaction between MET and mTOR, mTOR-related proteins. This value means a combined score with MET, mTOR, mTOR-related proteins, and its predicted target proteins in the STRING database. (**B**) HepG2 and Hep3B cells were transfected with IL-32γ (5 μg) for 24 h. The mRNA level of *mTOR*, *TP53*, *PIK3CA*, and *AKT1* was measured by RT-qPCR. Data are presented as mean ± S.D. from three independent experiments ***, *p* < 0.001 indicates statistically significant differences from the control group. (**D**) HepG2 and Hep3B cells were transfected with IL-32γ (0, 1, 2, and 5 μg) for 24 h. The expression of mTOR-associated proteins was determined by western blot analysis. (**E**) HepG2 and Hep3B cells were transfected with IL-32γ (2 μg) or Cabozantinib (2 μM) for 24 h. The expression of mTOR-related proteins was determined using western blotting.

**Figure 5 ijms-25-11678-f005:**
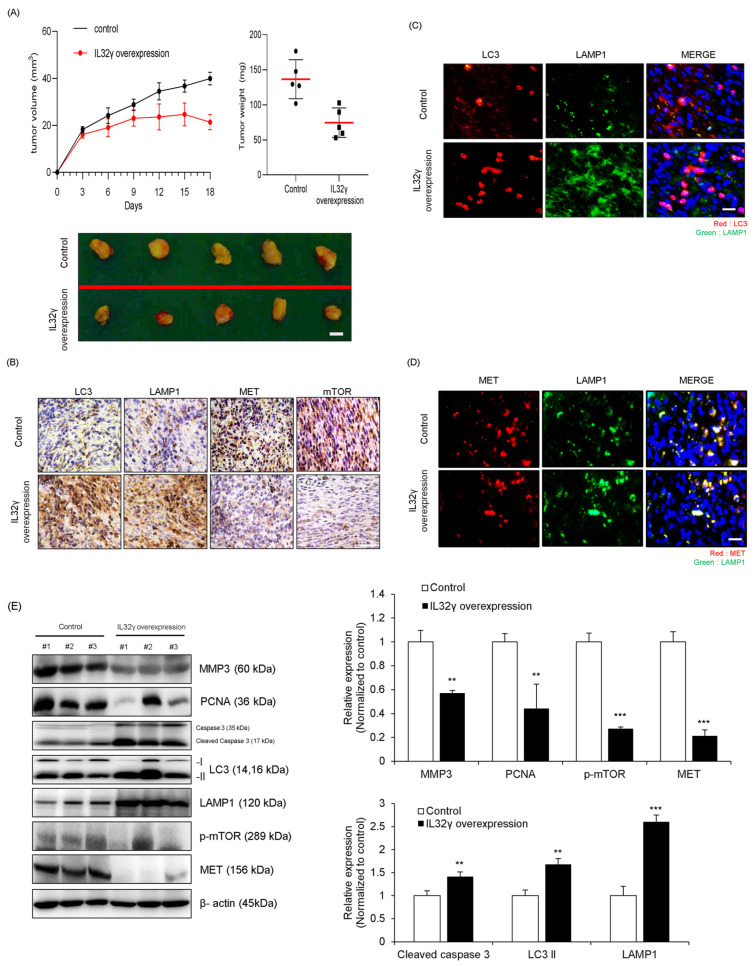
Effect of overexpression of IL-32γ on the induction of autophagy in vivo. HepG2 (1 × 10^7^ cells) was well injected subcutaneously to induce liver cancer tumors and checked twice a week for 3 weeks (*n* = 5 per group). (**A**) Representative image of tumors obtained from mice in each group. Scale bar, 0.5 cm. Tumor size and weight were measured, and tumor volume was calculated with the formula V = L × W2 × 0.5 (*n* = 5). ** *p* < 0.01; *** *p* < 0.001; (one-way ANOVA). (**B**) Representative immunohistochemistry images of tumor tissues in each group. Immunohistochemistry staining (LC3, LAMP1, p-mTOR, and MET) was repeated from three independent experiments. Scale bar: 50 μm. (**C**,**D**) Immunofluorescence analysis for LC3, LAMP1, and MET antibodies in control and IL-32γ overexpression mice tumor tissues. The sections were immunostained with LC3 (green), and LAMP1 (red), LAMP1 (green) and MET (red) during fusion with autophagosomes and lysosomes. The cell nucleus was stained with DAPI (blue). The data were the average of three independent experiments. Scale bar, 100 μm. (**E**) The mouse tissue extracts were subjected to western blot with indicated antibodies (MMP3, PCNA, Cleaved caspase 3, LC3, LAMP1, p-mTOR, and MET) and β-actin (internal control). **, *p* < 0.01, ***, *p* < 0.001 (one-way ANOVA).

**Figure 6 ijms-25-11678-f006:**
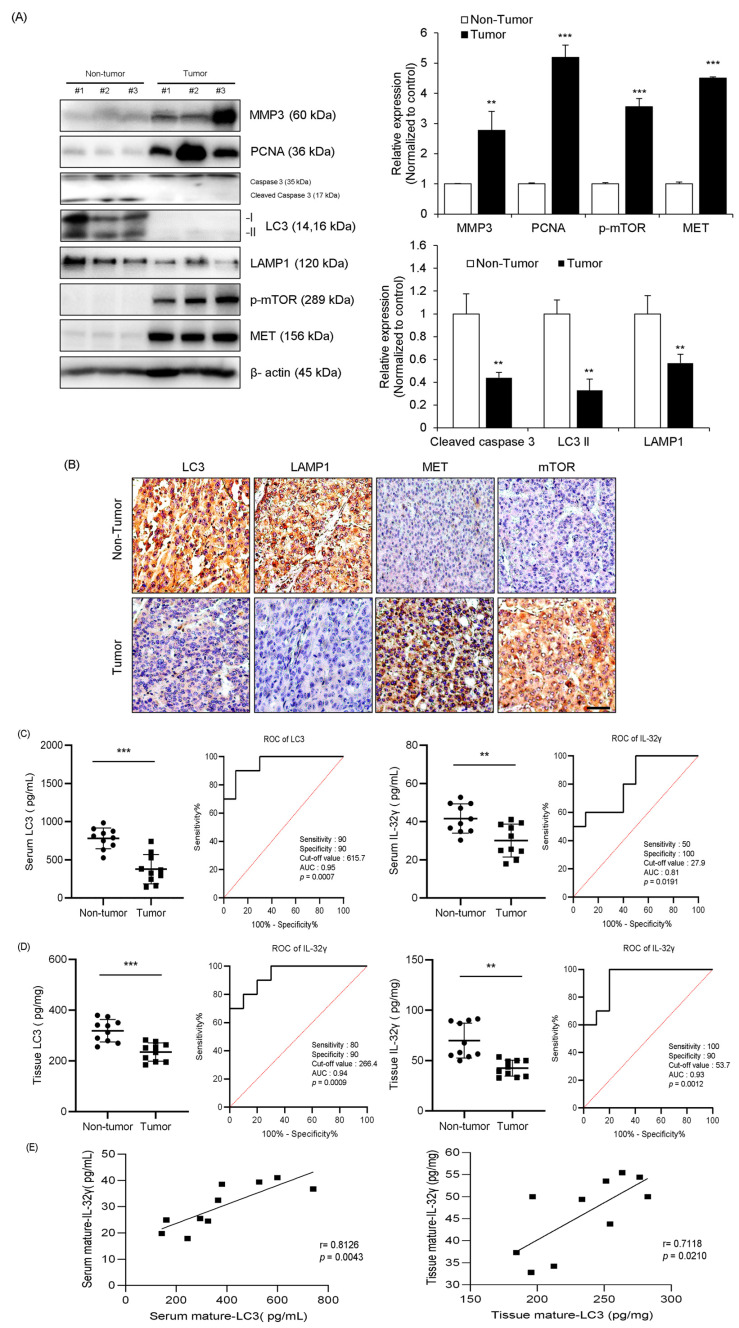
Autophagy and mTOR pathway play in human liver tumors of patients. (**A**) Autophagy-related markers (LC3 and LAMP1), MET, and p-mTOR expression in liver cancer patients and normals were analyzed by western blotting, *n* = 3 per group. The intensity of each band was measured, and the ratio of the amount of each protein to β-actin was calculated. Data are presented as mean ± standard deviation (SD) from three independent experiments. **, *p* < 0.01, ***, *p* < 0.001 (one-way ANOVA). (**B**) Representative images of tumor tissues from each group. Immunohistochemistry staining (LC3, LAMP1, p-mTOR, and MET) was repeated from three independent experiments. Scale bar: 100 μm. (**C**,**D**) Serum and tissue levels of LC3 and IL-32 γ in liver cancer patients and their control and receiver operating characteristic (ROC) curve. *n* = 10 for the control group; *n* = 10 for the patient group. *** *p* < 0.0012 (unpaired two-tailed *t*-test). (**E**) Spearman correlation test results between IL-32 and LC3.

**Table 1 ijms-25-11678-t001:** RT-qPCR primer list.

Gene	Direction	Sequence (5′ to 3′)	Gene ID
MET	Forward	5′-AGCAATGGGGAGTGTAAAGAGG-3′	ID 4233
	Reverse	5′-CCCAGTCTTGTACTCAGCAAC-3′	
PRKCD	Forward	5′-GTGCAGAAGAAGCCGACCAT-3′	ID 5580
	Reverse	5′-CCCGCATTAGCACAATCTGGA-3′	
IL18RAP	Forward	5′-ATGCTCTGTTTGGGCTGGATA-3′	ID 8807
	Reverse	5′-GTGAGAGTCGATTTCTGTGGC-3′	
CXCL8	Forward	5′-TTTTGCCAAGGAGTGCTAAAGA-3′	ID 3576
	Reverse	5′-AACCCTCTGCACCCAGTTTTC-3′	
TNF	Forward	5′-CCTCTCTCTAATCAGCCCTCTG-3′	ID 7124
	Reverse	5′-GAGGACCTGGGAGTAGATGAG-3′	
mTOR	Forward	5′-GCAGATTTGCCAACTATCTTCGG-3′	ID 2475
	Reverse	5′-CAGCGGTAAAAGTGTCCCCTG-3′	
TP53	Forward	5′-CAGCACATGACGGAGGTTGT-3′	ID 7157
	Reverse	5′-TCATCCAAATACTCCACACGC-3′	
AKT1	Forward	5′-GTCATCGAACGCACCTTCCAT-3′	ID 207
	Reverse	5′-AGCTTCAGGTACTCAAACTCGT-3′	
PIK3CA	Forward	5′-GAAACAAGACGACTTTGTGACCT-3′	ID 5290
	Reverse	5′-CTTCACGGTTGCCTACTGGT-3′	

## Data Availability

The original contributions presented in the study are included in the article. Further inquiries can be directed to the corresponding authors.

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
