# Peer review of "IL-32γ Induced Autophagy Through Suppression of MET and mTOR Pathways in Liver Tumor Growth Inhibition"

_ijms, 2024, doi:10.3390/ijms252111678_

Round 1
Reviewer 1 Report
Comments and Suggestions for Authors
In this study, the Authors investigated the role of IL-32γ in autophagy induction in liver cancer cells and tried to delineate the underlying mechanisms. They found that the increased IL-32γ expression inhibited the growth, cell cycle progression, and migration of HepG2 and Hep3B cell lines; it also decreased the expression of related proteins. Furthermore, the IL-32γ overexpression induced autophagy, as indicated by the number of puncta, the expression of LC3, and the expression of autophagy-related markers. The expression levels of LAMP1, a protein essential for autophagosome formation, and colocalization
with LC3 also increased. Big data analysis revealed that the expression of MET, a well-known target of autophagy, and the expression of mTOR and mTOR-related proteins were decreased by the IL-32γ overexpression. The combination treatment of MET inhibitor, cabozantinib (2 μM), and IL-32γ overexpression further increased the number of puncta, the colocalization of LC3 and LAMP1, and the expression of autophagy-related proteins. In vivo, liver tumor growth was suppressed in the IL-32γ-overexpressing mouse model, and autophagy induction was confirmed by the increased expression of LC3 and LAMP1 and the decreased expression of autophagy pathway markers (MET and mTOR). Autophagy was also decreased in the liver tumor sample of human patients. ROC curve and spearman analysis revealed that the expression levels of LC3 and IL-32γ were significantly correlated in human tumor serum and tissues. Therefore, IL-32γ overexpression induced autophagy in liver tumor through the suppression of MET and mTOR pathways critical for tumor growth inhibition.
Specific comments:
1. Microscopy images lack scale bars.
2. Western blot images are "squeezed" probably by unproportional formatting during figure preparation.
3. How can presented results be discussed in light of different basal autophagy levels in these two cell lines as can be deduced from the Fig. 2?
4. The results do not match between Fig. 2 and Fig. 3E when the same experimental points are compared.
5. The quantification of the Western blot data should be shown below the blots, not on the blots (for better clarity).
Overall, the manuscript shown valid and interesting results. However, the presentation of the results must be improved.
Reviewer 2 Report
Comments and Suggestions for Authors
Comments:
The manuscript describes "IL-32γ induced autophagy through suppression of MET and mTOR pathways in liver tumor growth inhibition”. Interleukin 32γ (IL-32γ) has multiple functions in various malignancies. After discovering the role of IL-32γ in the induction of autophagy in liver cancer cells, it was found that increased expression of IL-32γ inhibited the growth, cell cycle progression, and migration of HepG2 and Hep3B cell lines; it also reduced the expression of related proteins. The analysis showed that IL-32γ overexpression reduces the expression of MET, a well-known autophagy target, and the expression of mTOR and mTOR-related proteins. Combined treatment with the MET inhibitor cabozantinib and IL-32γ overexpression further increased the co-localization of LC3 and LAMP1 and the expression of autophagy-related proteins. In vivo, liver tumor growth was inhibited in an IL-32γ overexpressing mouse model, and autophagy induction was confirmed by increased expression of LC3 and LAMP1 and decreased autophagy pathway markers (MET and mTOR). Therefore, IL-32γ overexpression may induce autophagy in liver tumors by inhibiting the MET and mTOR pathways that are critical for suppressing tumor growth., but several points need clarification.
Comment:
1. In the figure, WB analysis requires quantitative analysis and statistics.
2. Excessive expression of IL-32γ can induce autophagy in liver tumors by inhibiting tumor growth and the MET and mTOR pathways. The authors should add a mechanism of action to facilitate readers.
3. The resolution of the chart is very poor. The author should improve it to make it easier for readers to read.
Round 2
Reviewer 1 Report
Comments and Suggestions for Authors
Previous comments have been addressed.
Please only correct line 103 - there shoudl be IL-32y, not L32y.
Reviewer 2 Report
Comments and Suggestions for Authors
Accepted